# Comparative Efficacy and Safety of Anti-PD-1/PD-L1 Immune Checkpoint Inhibitors for Refractory or Relapsed Advanced Non-Small-Cell Lung Cancer—A Systematic Review and Network Meta-Analysis

**DOI:** 10.3390/cancers13010052

**Published:** 2020-12-27

**Authors:** Koichi Ando, Ryo Manabe, Yasunari Kishino, Sojiro Kusumoto, Toshimitsu Yamaoka, Akihiko Tanaka, Tohru Ohmori, Tsukasa Ohnishi, Hironori Sagara

**Affiliations:** 1Division of Respiratory Medicine and Allergology, Department of Medicine, Showa University School of Medicine, 1-5-8 Hatanodai, Shinagawa-ku, Tokyo 142-8666, Japan; r.manabe@med.showa-u.ac.jp (R.M.); ookiyookiy@med.showa-u.ac.jp (Y.K.); k-sojiro@med.showa-u.ac.jp (S.K.); tanakaa@med.showa-u.ac.jp (A.T.); tohmorit@med.showa-u.ac.jp (T.O.); tohnishi@med.showa-u.ac.jp (T.O.); sagarah@med.showa-u.ac.jp (H.S.); 2Division of Internal Medicine, Showa University Dental Hospital Medical Clinic, Showa University Senzoku Campus, 2-1-1 Kita-senzoku, Ohta-ku, Tokyo 145-8515, Japan; 3Advanced Cancer Translational Research Institute (Formerly, Institute of Molecular Oncology), Showa University, 1-5-8 Hatanodai, Shinagawa-ku, Tokyo 142-8555, Japan; yamaoka.t@med.showa-u.ac.jp

**Keywords:** non-small-cell lung cancer, immune checkpoint inhibitors, overall survival, progression-free survival, network meta-analysis, systematic review

## Abstract

**Simple Summary:**

Improving treatment strategies for refractory or relapsed advanced non-small-cell lung cancer (NSCLC) remains a challenge. Efficacy and safety of the immune checkpoint inhibitors (ICIs), nivolumab (Niv) plus atezolizumab (Atz), were compared with those of ramucirumab (Ram) plus docetaxel (Doc), and the efficacy and safety of these two ICIs were compared with each other, using patient groups without programmed cell death ligand-1 (PD-L1) constraint. Additionally, efficacy and safety of, Niv, Atz, and pembrolizumab (Pem) were compared using a PD-L1 positive (≥1%) subgroup with refractory or relapsed advanced NSCLC. Niv or Atz was found to be more effective and safer than Ram plus Doc in groups without PD-L1 constraint. In the PD-L1 positive subgroup, Pem (10 mg/kg) showed the highest efficacy for ensuring overall survival, followed by Niv, Pem (2 mg/kg), Atz, and Doc. These results may help clinicians select and evaluate treatment options for relapsed or refractory advanced NSCLC.

**Abstract:**

The efficacy and safety of immune checkpoint inhibitors (ICIs) in refractory or relapsed advanced non-small-cell lung cancer (NSCLC) have not yet been compared with those of ramucirumab (Ram) plus docetaxel (Doc). Furthermore, comprehensive comparisons between ICIs have not been conducted to date. In the current study, a Bayesian network meta-analysis of related phase III clinical trials was performed to compare the efficacy and safety of Ram+Doc, Niv, Atz, and Doc treatments in patient groups lacking the PD-L1 constraint. Surface under the cumulative ranking area (SUCRA) revealed that the overall survival (OS) of patients treated with Niv was the highest, followed by Atz, Ram+Doc, and Doc. Regarding grades 3–5 treatment-related adverse events (G3–5AEs), the use of Niv was ranked the safest, followed by Atz, Doc, and Ram+Doc. Significant differences in OS were observed between Niv and Ram+Doc, while significant differences in G3–5AEs were observed between Ram+Doc and Niv or Atz. In the PD-L1 positive (≥1%) patient subgroup, Pem (10 mg/kg) ranked the highest in efficacy for OS, followed by Niv, Pem (2 mg/kg), Atz, and Doc. These findings may expectedly provide oncologists with useful insights into therapeutic selection for refractory or relapsed advanced NSCLC.

## 1. Introduction

Advances in molecular and immunologic chemotherapies [1,2,3] have resulted in the development of novel strategies for lung cancer treatment [1,2,3,4]. Despite these advances, lung cancer remains a major source of cancer-related death, accounting for 13% of all cancer mortalities, which is attributed to the poor 5-year survival rate (18%) associated with this disease [5]. When detected at an advanced stage, as is common in the more prevalent (84%) non-small-cell lung cancer (NSCLC) cases, the 5-year survival rate is only 5% [2,3,5]. Furthermore, relapse is a common occurrence in NSCLC, even following a positive response to initial treatment. Although these findings are based on data obtained from patients who had not been treated with immunotherapy, the prognosis for advanced NSCLC remains mostly poor. Therefore, improved and modified treatment strategies are warranted for refractory or relapsed NSCLC [1,2,3,5,6,7,8].

Recently, the efficacy of a combination of Ram and Doc (Ram+Doc) was evaluated as a second-line treatment for advanced NSCLC [9]. Ram+Doc treatment significantly improved overall survival (OS) as well as progression-free survival (PFS) in comparison with Doc monotherapy. Consequently, the Ram+Doc combination remains the treatment of choice for driver mutation-negative refractory or relapsed advanced NSCLC, or cancer of unknown status [2,3,9,10]. Nintedanib plus Doc, a treatment that has been approved based on the results of previous phase III trials, may present a new treatment option for adenocarcinoma [11].

Immunotherapy, including the use of immune checkpoint inhibitors (ICIs), has recently been approved by the FDA for lung cancer treatment and is widely used for this purpose [1,4,12,13]. Treatment with Niv, Atz, and Pem, in particular, have significantly improved the outcome of NSCLC patients [1,3,4,5,12,13].

However, to date, only a few studies have compared ICIs with Ram+Doc treatments. Moreover, studies comparing the efficacy and safety profiles of ICIs with each other in refractory or recurrent advanced NSCLC are scant. Although ICIs have been approved for the treatment of refractory or relapsed NSCLC [1,6], there is limited evidence suggesting that ICIs are superior to conventional therapies, such as Ram+Doc. Furthermore, currently, there is insufficient evidence indicating which ICIs are the most effective as well as safe for use in refractory or recurrent advanced NSCLC. Thus, a comparative analysis of ICIs and Ram+Doc treatments leading to a ranking in the order of usefulness or efficacy against this disease may be warranted.

In general, randomized controlled trials (RCTs) are considered to be the most appropriate for comparing the efficacy and safety of treatment interventions. However, RCTs are beset by limitations such as time constraints as well as logistical and financial considerations. Therefore, we adopted another statistical method, network meta-analysis (NMA), as an alternative [14,15,16]. Our study design permits the efficacy and safety of any treatment pair to be compared and ranked more rapidly than it would be possible via RCTs, even in the absence of previous studies of directly comparative RCTs [14,15,16,17,18,19,20,21,22]. Thus, using this systematic review and meta-analysis (registration: UMIN-CTR no. UMIN000041086), we compared and ranked the efficacy and safety of ICIs, Doc and Ram+Doc in treating refractory or relapsed patients with NSCLC using statistical Bayesian NMA. Our results may expectedly provide insights into the most efficacious and safe treatments for NSCLC.

## 2. Results

### 2.1. Systematic Review

Of the 1526 articles identified via a systematic literature review that met the search criteria (121 from PubMed [23], 276 from Embase [24], 345 from the Cochrane Central Register of Controlled Trials (CENTRAL) [25], and 784 from SCOPUS [26]), 1084 remained following the removal of duplicates (Figure 1). The application of the collation of Patients, Intervention, Comparison, Outcome and Study (PICOS) criteria (Table 1) suggested that four studies were in order: one comparing Ram+Doc with Doc [9]; and the others comparing Doc with Niv [27,28] or Atz [29]; (Table 2).

The common comparative group in these four studies was the Doc group [9,27,28,29]. The data obtained from these studies were sufficient to perform an NMA for OS, including a subgroup analysis based on histology. However, these data were not sufficient to conduct a subgroup analysis of PFS. Therefore, PFS was analyzed only in the overall patient group. In all analyses, the preferred model convergence was confirmed using the Brooks–Gelman–Rubin method [30,31]. 

Although Pem is indicated for the treatment of NSCLC, immunological regimens comprising Pem were not included in the present NMA owing to the heterogeneity of target patients. Phase III trials, in which refractory or relapsed NSCLC was treated with Pem [32], included a patient group with limited PD-L1 expression, which was considered unsuitable for inclusion in this NMA. By contrast, CheckMate057, CheckMate017, and OAK, like REVEL, did not place a limit on PD-L1 positivity in their patient inclusion criteria. Therefore, including these four RCTs in the analysis allowed each treatment group to be compared in a uniform PD-L1 status environment. The network map of the NMA conducted in the present study is shown (Figure 2).

### 2.2. Primary Efficacy Endpoint: OS

The OS of patients treated with Atz, Niv, and Ram+Doc was significantly higher than that of patients treated with Doc (hazard ratio [HR] [95% credible intervals {CrIs}]: 0.732 [0.615–0.864], 0.683 [0.575–0.806], and 0.862 [0.752–0.983], respectively). The OS of patients in the Niv group was significantly higher than that of patients in the Ram+Doc group (HR: 0.796 [95% CrI: 0.639–0.982]); however, there were no significant differences in the OS of patients receiving Atz vs. Ram+Doc (HR: 0.854 [95% CrI: 0.683–1.054]) or those receiving Atz vs. Niv (HR: 1.080 [95% CrI: 0.843–1.360]) (Figure 3).

### 2.3. Subgroup Analysis of the OS of Patients with Non-Squamous NSCLC

The OS of patients in the Atz, Niv and Ram+Doc groups was significantly higher than that of patients in the Doc group (HRs [95% CrIs]: 0.733 [0.598–0.888], 0.734 [0.595–0.898], and 0.833 [0.708–0.972], respectively). However, no statistically significant differences were noted for the Atz vs. Niv groups, Atz vs. Ram+Doc groups, and Niv vs. Ram+Doc groups (HRs [95% CrIs]: 1.010 [0.750–1.327], 0.887 [0.682–1.133], and 0.888 [0.679–1.142], respectively) (Figure 4). 

### 2.4. Subgroup Analysis of OS of Patients with Squamous NSCLC

The OS of patients in the Atz and Niv groups was significantly higher than that of patients in the Doc group (HRs [95% CrIs]: 0.738 [0.540–0.982] and 0.597 [0.441–0.793], respectively). The OS of patients in the Niv group was improved compared with that of patients in the Ram+Doc group (HR: 0.682 [95% CrI: 0.457–0.981]). By contrast, no statistically significant differences were noted for Atz vs. Niv, Atz vs. Ram+Doc or Ram+Doc vs. Doc (HRs [95% CrIs]: 1.264 [0.812–1.874], 0.843 [0.561–1.216], and 0.890 [0.691–1.126], respectively) (Figure 5).

### 2.5. Secondary Efficacy Endpoint: PFS

The PFS of patients in the Niv and Ram+Doc groups was significantly higher than that of patients in the Doc group (HRs [95% CrIs]: 0.817 [0.700–0.948] and 0.761 [0.676–0.855], respectively), whereas no significant differences were observed in PFS for Atz vs. Doc (HR: 0.952 [95% CrI: 0.819–1.099]). Atz was significantly inferior to Ram+Doc in terms of PFS (HR: 1.255 [95% CrI: 1.035–1.509]), whereas no significant differences were noted for the Atz vs. Niv or Niv vs. Ram+Doc (HRs [95% CrIs]: 1.173 [0.943–1.440] and 1.076 [0.885–1.299], respectively) (Figure 6).

### 2.6. Primary Safety Endpoint: Grade 3–5 Treatment-Related Adverse Events (G3–5AEs)

G3–5AEs were significantly more common in the Ram+Doc group than in the Doc group (risk ratio [RR]: 1.101; 95% CrI: 1.032–1.172). The incidence of G3–5AEs in the Atz and Niv groups was significantly lower than that of the Doc group (RRs [95% CrIs]: 0.342 [0.274–0.421] and 0.175 [0.127–0.235], respectively). The incidence of G3–5AEs in the Atz and Niv groups was significantly lower than that of the Ram+Doc group (RRs [95% CrIs]: 0.311 [0.247–0.386] and 0.159 [0.115–0.215], respectively). The incidence of G3–5AEs in the Atz group was significantly higher than that of the Niv group (RR: 2.005 [95% CrI: 1.350–2.862]) (Figure 7).

### 2.7. Ranking Assessment

The efficacy and safety of Ram+Doc, Doc, Niv, and Atz treatments were ranked according to the surface under the cumulative ranking area (SUCRA) curve. Higher SUCRA values indicate better outcomes. SUCRA values were estimated for efficacy and safety outcome indicators, OS, PFS, and G3–5AEs, while a subgroup analysis for OS was conducted based on histological data. SUCRA analyses indicated that OS was highest for Niv (SUCRA, 90.0%) followed by Atz (SUCRA, 73.8%), Ram+Doc (SUCRA, 35.7%), and Doc (SUCRA, 0.5%); (Figure 8A).

SUCRA values for the OS of non-squamous NSCLC patients indicated that Atz treatment (78.1%) was the best, followed by Niv (77.7%), Ram+Doc (43.8%), and Doc (0.4%); (Figure 8B). The SUCRA values for OS of squamous NSCLC patients indicated that Niv treatment (94.0%) was the best, followed by Atz (65.9%), Ram+Doc (34.2%), and Doc (6.0%); (Figure 8C). The SUCRAs for PFS indicated that Ram+Doc treatment (91.6%) was the best, followed by Niv (72.0%), Atz (28.0%), and Doc (8.3%); (Appendix A). The SUCRAs for G3–5AEs indicated that Niv treatment (100.0%) was the best, followed by Atz (66.7%), Doc (33.3%), and Ram+Doc (0.1%) treatments (Figure 8A–C and Appendix A).

### 2.8. Sensitivity Analysis

The OAK study included a group of patients who had received one previous systemic treatment and another group of patients who had received two systemic treatments. By contrast, other studies, such as CheckMate017, CheckMate057, and REVEL, included only patients who had received one previous systemic treatment.

To evaluate the effect of including or excluding the patient group that received two prior systemic treatments in the OAK study, we performed a sensitivity analysis for the primary endpoint (OS). The sensitivity analysis of almost any paired-treatment comparison designed to determine statistical significance presented a similar result. Furthermore, there was negligible change in the effect size and SUCRA values, and the ranking of the primary efficacy outcome was similar for each treatment. The results of the sensitivity analyses are presented (Appendix A). Based on these results, we considered that the inclusion or exclusion of patients with two prior systemic treatments in OAK did not affect our conclusions.

### 2.9. Bias Assessment

The quality of the studies included in the analysis was assessed using the Cochrane-recommended risk of bias tool 2 (ROB2) [33]. Although three studies presented some concerns regarding the bias attributed to deviation from intended intervention, due to their open-label design, the remaining study was considered to have a low risk of bias. Other parameters evaluated, i.e., bias arising from the randomization process, bias due to missing outcome data, bias in measurement of the outcome and bias in the selection of the reported results, were judged to be low risk in all four studies; thus, the quality of the included studies was considered to be good (Appendix A).

### 2.10. Comparison between the Effect of ICIs on Refractory or Relapse PD-L1-Positive (≥1%) Advanced NSCLC

#### 2.10.1. Network Meta-Analysis of Six RCTs from REVEL, CheckMate057, CheckMate017, OAK, LUME-lung 1 and KEYNOTE-010, for Predefined Efficacy and Safety Outcomes (OS, PFS, and G3-5AEs)

Prior to comparing ICIs in the PD-L1-positive (≥1%) group, we performed a network meta-analysis of predefined efficacy and safety outcomes by including the overall participants of six studies, REVEL, CheckMate057, CheckMate017, OAK, LUME-lung 1 and KEYNOTE-010, and based on its results, conducted a subgroup analysis of refractory or relapse PD-L1-positive (≥1%) advanced NSCLC. Although KEYNOTE-010 and LUME-Lung 1 did not satisfy the predetermined criteria for inclusion in the NMA for a patient group lacking PD-L1 constraint, we surmised that the additional inclusion of Pem and Nin+Doc into the treatment arms of NMA may lead to further interesting findings. Based on this contention, we attempted to assess predefined efficacy and safety outcomes using these two trials, in addition to the four already included, for a total of six trials (REVEL, CheckMate057, CheckMate017, OAK, LUME-Lung 1, KEYNOTE-010). 

The key inclusion criteria, main characteristics and results of risk of bias assessment of LUME-Lung 1, a phase III study of Nin+Doc, and KEYNOTE-010, a Phase II/III study of Pem, are shown (Appendix A, and Appendix A, respectively). 

A comparative analysis which compared the effect of the seven treatments, Doc, Ram+Doc, Niv, Atz, pem 2 mg/kg (Pem2), pem 10 mg/kg (Pem10) and Nin+Doc, on the OS of the overall participants of six RCTs, REVEL, CheckMate057, CheckMate017, OAK, KEYNOTE-010, and LUME-Lung 1, was performed. The network map is shown in Figure 9, and the results are shown in Appendix A.

Furthermore, five RCTs, REVEL, CheckMate057, OAK, KEYNOTE-010 and LUME-Lung 1, were also included in the analysis that compared the effects of the six treatments, Doc, Ram+Doc, Niv, Atz, pem 2 mg/kg or 10 mg/kg (Pem pooled), and Nin+Doc on the OS of the non-squamous subgroup (Appendix A). In addition, four RCTs, REVEL, CheckMate017, OAK and KEYNOTE-010, were included in the analysis that compared the efficacy of the five treatments, Doc, Ram+Doc, Niv, Atz and Pem (pooled), on the OS of the squamous subgroup, (Appendix A). A further analysis compared the effects of the seven treatments, Doc, Ram+Doc, Niv, Atz, Pem2, Pem10, and Nin+Doc, on the PFS for overall participants of six RCTs, REVEL, CheckMate057, CheckMate017, OAK, KEYNOTE-010, and LUME-Lung 1. (Appendix A). In addition, another analysis compared the effects of seven treatments, Doc, Ram+Doc, Niv, Atz, Pem2, Pem10 and Nin+Doc, on safety, as indicated by G3-5AEs, of the overall participants of six RCTs, REVEL, CheckMate057, CheckMate017, OAK, KEYNOTE-010 and LUME-Lung 1 (Appendix A). 

The results of ranking assessment as indicated by SUCRA value are presented (Appendix A).

#### 2.10.2. Subgroup Analysis for Refractory or Relapse PD-L1-Positive (≥1%) Advanced NSCLC

We performed a subgroup analysis for the PD-L1-positive (≥1%) patient population and compared the efficacy for OS between ICIs. Four RCTs, CheckMate057, CheckMate017, OAK, and KEYNOTE-010, were included in this subgroup analysis which compared the efficacy of five treatments, Doc, Niv, Atz, Pem2 and Pem10. The network map of the NMA conducted as the subgroup analysis for refractory or relapse PD-L1-positive (≥1%) advanced NSCLC is shown in Figure 10, and the results are shown in Figure 11.

Furthermore, in the analysis of the PD-L1-positive (≥1%) patient population, we added a subgroup analysis by histological type (non-squamous subgroup or squamous subgroup). The results are shown (Appendix A). 

The results of ranking assessment by using SUCRA values for efficacy in OS in the PD-L1-positive group and its subgroups by histology (non-squamous or squamous) are shown (Appendix A). 

## 3. Discussion

The current NMA compared the efficacy and safety of ICIs with Ram+Doc for refractory or relapsed advanced NSCLC in patient groups lacking the PD-L1 status constraint. The primary efficacy endpoint, OS, was significantly improved following treatment with Niv compared with Ram+Doc treatment. However, no significant differences were observed between the Atz and Ram+Doc groups. The incidence of G3–5AEs was lower in both Niv and Atz groups compared with that of the Ram+Doc group. The Niv group displayed the highest SUCRA for OS, followed by Atz, Ram+Doc and Doc groups. The Niv group showed the highest SUCRA for G3–5AEs (lowest incidence of G3–5AEs), followed by Atz, Doc and Ram+Doc groups. In subgroup analysis of the PD-L1-positive (≥1%) patient population, ICI-to-ICI comparisons indicated that Pem10 displayed the highest efficacy for OS, followed by Niv, Pem2 and Atz, although there were no significant differences between these ICIs.

Previous studies have compared the efficacy and safety of Ram+Doc and Doc [9], Niv and Doc [27,28], Atz and Doc [29], and Pem and Doc in cases limited to PD-L1 (≥1%) positive patients [32]. In patients with refractory or relapsed advanced NSCLC, Ram+Doc, Niv, Atz, and Pem significantly improved OS compared with Doc [9,27,28,29,32]. However, no studies have compared the efficacy and safety of ICIs with those of Ram+Doc. Moreover, there are no reports comparing the efficacy and safety of ICIs. In this systematic review and meta-analysis, we adopted a unique approach which simultaneously compared and ranked the efficacy and safety of four treatments, Ram+Doc, Doc, Niv and Atz, which are indicated for relapsed or refractory NSCLC in patients lacking the PD-L1 constraint. We also compared the inter-ICI efficacy of Niv, Atz and Pem for affecting OS in the PD-L1-positive (≥1%) patient subgroup. The results of the present study suggest that Niv and Atz displayed superior efficacy and safety profiles than did Ram+Doc for relapsed or refractory advanced NSCLC in patients lacking the PD-L1 constraint. This is an important clinical finding that will enhance the ability of oncologists to treat patients more effectively. Our results showed that ICIs displayed favorable efficacy and safety profiles compared with those of Ram+Doc for relapsed or refractory advanced NSCLC. Although Pem could not be included in the treatment comparison analysis for patient groups lacking the PD-L1 constraint, due to non-compliance with our predetermined inclusion criteria, a comparison between Niv and Atz indicated that Niv showed relatively better efficacy and safety profiles than did Atz. Particularly, with respect to the safety profile, Atz had a significantly higher frequency of G3–5AEs than did Niv. In the subgroup analysis of the PD-L1-positive (≥1%) patient population, Pem10 had the best efficacy profile, followed by Niv, Pem2, Atz, and finally Doc, although the results did not show a significant difference between these. Furthermore, PD-L1-positive (≥1%) cases were analyzed by histological type, allowing comparison between Pem (pooled) and Niv. The results showed that no significant differences existed between the effects of Pem (pooled) and Niv on OS in non-squamous and squamous patients, respectively.

The molecular basis underlying these findings requires clarification: Cell death caused by systemic anticancer therapy, such as chemotherapy, elicits an inflammatory response that induces the accumulation of activated T cells. Activated T cells are suppressed by cancer cells via PD-L1, which is eliminated by treatment with Niv and Atz, during secondary and subsequent therapies, thereby restoring the anticancer activity of T cells as well as apoptosis of cancer cells [1,4,12,34]; (Figure 12). These findings explain the process by which treatment response in advanced NSCLC is improved by ICIs, Niv and Atz, in comparison to chemotherapy (Doc), with or without Ram, during second-line and subsequent treatments.

However, there are reported cases of primary or adaptive resistance to cancer immunotherapeutic agents such as Niv [1,4,6]. Moreover, some patients who are resistant to immunotherapy have been reported to respond to a treatment regimen that includes Ram [10]. These reports not only suggest the importance of appropriate therapeutic selection for each patient, but also the importance of conducting a detailed investigation of patient characteristics that are most likely to benefit from each therapeutic agent. 

Few reports have evaluated the efficacy of Pem in a refractory or relapsed advanced PD-L1-negative (<1%) NSCLC patient group. Therefore, we were unable to include Pem in an analysis of a patient group lacking the PD-L1 constraint. Although an efficacy analysis was performed on the whole patient population, including Pem, the results of which are shown (Appendix A), these results must be interpreted with caution on account of the heterogeneity of PD-L1 status. Further validation of the efficacy and safety profile of Pem in a refractory or relapsed PD-L1-negative (<1%) advanced NSCLC group of patients may be necessary.

In an inclusive efficacy analysis of the six RCTs (REVEL, CheckMate057, CheckMate017, OAK, LUME-Lung 1, KEYNOTE-010), conceptual heterogeneity existed in patient inclusion criteria for PD-L1 status in each RCT. That is, while only PD-L1-positive (≥1%) patients were included in KEYNOTE-010, PD-L1 status was not taken into consideration in patient inclusion for other RCTs. This may be considered as a limitation when comparing ICIs. By contrast, comparisons between Ram+Doc and Nin+Doc were considered to be feasible. Our analysis showed that there was no statistically significant difference in OS between Ram+Doc and Nin+Doc in the non-squamous subgroup.

Although only Niv and Atz were included and Pem was excluded, in the comparison of ICIs in the patient group lacking the PD-L1 constraint, the results led to some interesting observations, especially regarding efficacy outcomes. Subgroup analysis by histological type showed that the highest OS for a drug differed according to histological type as follows: in squamous cell carcinoma, Niv showed the highest OS, whereas in non-squamous cell carcinoma, Atz showed the highest OS. Hence, although safety outcomes cannot be analyzed via histology, these results suggested that histology-specific treatment strategies should be considered for recurrent NSCLC. However, direct comparative RCTs may be required to confirm our results.

Analyses of safety profiles that were performed using the six RCTs, REVEL, CheckMate057, CheckMate017, OAK, LUME-Lung 1, and KEYNOTE-010, allowed the incidence of G3-5AEs among Doc, Ram+Doc, Niv, Atz, Pem and Nin+Doc to be compared. The results indicated that Niv was the safest, followed by Atz, Pem2, Pem10, Doc, Ram+Doc, and Nin+Doc (Appendix A). The differences between Ram+Doc and Doc, Niv and Doc, Atz and Doc, Pem2 and Doc Pem10 and Doc, Nin+Doc and Doc, Niv and Ram+Doc, Atz and Ram+Doc, Pem2 and Ram+Doc, Pem10 and Ram+Doc, Atz and Niv, Pem2 and Niv, Pem10 and Niv, Nin+Doc and Niv, Nin+Doc and Atz, Nin+Doc and Pem2, and Nin+Doc and Pem10 were found to be statistically significant (Appendix A). The results of this safety profile analysis also provided clinically relevant information for the selection of therapeutic agents for refractory or relapsed advanced NSCLC.

Several limitations were associated with this study. As this was an NMA of RCTs that were performed separately, inconsistencies between these RCTs were not addressed in this study. Secondly, the inclusion criteria for each study differed. For example, there was a slight difference in the classification of patient stages; only stage IV patients were included in the REVEL trial, whereas stage IIIB and IV patients were included in the remaining three trials (CheckMate057, CheckMate017, and OAK). Furthermore, the number of prior treatments also differed; the OAK trial included one or two prior systemic cancer treatments, whereas the remaining three trials (REVEL, CheckMate017, and CheckMate057) included only one systemic cancer treatment. To address this limitation, we performed a sensitivity analysis which confirmed that the inclusion/exclusion of patients who had received two prior treatments had little effect on the final conclusions of this NMA. Nevertheless, this heterogeneity cannot be completely ignored since the sensitivity analysis cannot eliminate all effects of heterogeneity on the results as well as conclusions. Thirdly, the efficacy profile of Pem in a population that included a PD-L1-negative (<1%) group was not evaluated in sufficient detail in the present NMA. This was due to the paucity of existing RCTs of Pem in the patient groups including refractory or recurrent PD-L1-negative (<1%) NSCLC patients. Therefore, the efficacy profile of Pem in a patient population including a group of patients with PD-L1-negative refractory or relapsed advanced NSCLC may need to be investigated in more detail in the future. Finally, the group of patients included in this analysis received platinum-based chemotherapy as the first-line treatment; these patients did not receive the current standard of care, that is, a combination of ICIs and platinum-based chemotherapy. Therefore, the results of this analysis are applicable only to a group of patients who received platinum-based chemotherapy without ICIs as the first-line treatment. 

## 4. Materials and Methods 

### 4.1. Systematic Review

Four databases, PubMed, CENTRAL, Embase, and SCOPUS, were searched in July 2020 for reported RCTs of ICIs or Ram+Doc for advanced NSCLC from 1946 onward. Our search strategy used keywords such as NSCLC, ICIs, ramucirumab, docetaxel, and their Medical Subject Headings in only English articles (Appendix B). All reference lists were reviewed to prevent any relevant studies from being missed.

The present study was conducted in accordance with the Preferred Reporting Items for Systematic Review and Meta-Analysis (PRISMA) [35] and the PRISMA extension statement for network meta-analysis [21]. The literature search was performed independently by two investigators (K.A. and Y.K.); in cases of conflict, a third researcher (T.Y.) participated for the purpose of conflict resolution. For the studies retrieved via a systematic review, inclusion and exclusion criteria were adapted using the PICOS approach to address clinical or methodological inter-study heterogeneity and ensure validity of the NMA.

### 4.2. Quality Evaluation

We evaluated the qualities of RCTs included in the present NMA using ROB2 recommended by the Cochrane Collaboration [33]. We assessed the following parameters as having a low risk, some concerns, or a high risk: (1) bias arising from the randomization process, (2) bias due to deviations from intended interventions, (3) bias due to missing outcome data, (4) bias in measuring the outcome and (5) bias in selection of the reported result. Quality evaluation was performed independently by two investigators (K.A. and S.K.), where in cases of conflict, a third investigator (T.Y.) was consulted for the purpose of conflict resolution.

### 4.3. Inclusion and Exclusion Criteria (Predefined PICOS)

#### 4.3.1. Patients

The following inclusion criteria were used for this analysis: age over 18, histological or cytological confirmation of refractory or relapsed NSCLC and performance status of 0 to 2 (on a 5-point scale, with a higher number indicating poorer general condition).

#### 4.3.2. Interventions/Comparisons

For the meta-analysis, we included patients that received at least one of the following treatments: (1) 3 mg/kg of Niv every 2 weeks, (2) 1200 mg/body of Atz every 3 weeks and (3) 10 mg/kg Ram every 3 weeks plus 75 mg Doc per square meter of body surface area every 3 weeks. The above dosages were selected based on licensed dosages and those reported in phase III trials. Doc was considered as the common comparative group since it was the standard treatment regimen prior to the development of immunotherapy and molecular cancer therapies for refractory or recurrent advanced NSCLC [3].

#### 4.3.3. Outcomes

The primary and secondary efficacy outcomes were OS and PFS, respectively, which were expressed using the HRs and associated 95% CrIs. The primary safety endpoint was G3–5AEs, which were expressed using RRs and associated 95% CrIs. To assess efficacy rankings, we calculated SUCRAs for OS and PFS in the overall population and in the subgroups based on histological types (non-squamous and squamous). In order to assess safety rankings, we calculated SUCRAs for the frequency of G3–5AEs. These analyses were performed only if they were obtained from studies that included data required for the analysis.

#### 4.3.4. Study Design

Inclusion criteria for the present NMA comprised randomized, parallel group, phase III studies. 

### 4.4. Statistical Method of NMA

We performed Bayesian NMA according to a rigorously constructed framework established by the National Institute for Health and Care Excellence [14,15,16,17,18,20,31,36]. These formulations have also been applied in various other clinical studies [37,38,39,40,41,42,43]. In this study, we used the standard Bayesian model described by Dias et al. [44,45,46], which assumes inconsistency and heterogeneity across all included studies [19]. The posterior distribution of the treatment effect was estimated by applying a non-informative prior distribution and Gibbs sampling based on the Markov chain Monte Carlo method. The number of iterations was set at 50,000, and the first 10,000 iterations were considered as a burn-in sample to exclude the effects of the initial values. Effect sizes were expressed as HR and RR along with associated 95% CrIs; if the 95% CrI of a HR or RR did not include 1, the difference between treatment groups was considered statistically significant. To estimate the ranking of each treatment effect, SUCRA was calculated. SUCRA values ranged from 0% to 100%, with higher SUCRA values indicating a relatively more favorable treatment [22]. We used the Brooks–Gelman–Rubin (BGR) diagnostic approach [30,31] to conduct convergence diagnoses of all comparisons. Both visual assessments and diagnoses used to confirm model convergence were conducted using the BGR method. OpenBUGS 1.4.0 (MRC Biostatistics Unit, Cambridge Public Health Research Institute, Cambridge, UK) was used for analysis and STATA (ver. 14; StataCorp, Cambridge, UK) was used to graphically display the results (College Station, TX, USA).

### 4.5. Ethical Aspects

Owing to the characteristics of the review and meta-analysis carried out in the present study, the need for institutional review board approval and informed consent was waived.

## 5. Conclusions

The Bayesian NMA comparing ICIs and Ram+Doc for refractory or relapsed advanced NSCLC in patient groups lacking the PD-L1 status constraint revealed that the OS corresponding to Niv treatment was superior to that of Ram+Doc. G3–5AEs were more common after treatment with Ram+Doc than after Atz or Niv. SUCRA values indicated that Niv displayed the highest OS and G3–5AEs. Our results indicated that treatment of refractory or relapsed advanced NSCLC with either Atz or Niv monotherapy was more effective than treatment with Doc or Ram+Doc for patients lacking the PD-L1 status constraint. Regarding PD-L1-positive (≥1%) refractory or relapsed advanced NSCLC subgroup, Pem10 was the most effective treatment, followed by Niv, Pem2, Atz, and finally Doc. Our findings may lead to more precise clinical decision-making in the future. However, verification using direct head-to-head RCTs seems to be necessary. Furthermore, characterizing the patient profiles that benefit the most from each treatment agent prescribed for refractory or relapsed advanced NSCLC is an important topic that needs to be addressed via future clinical research.

## Figures and Tables

**Figure 1 cancers-13-00052-f001:**
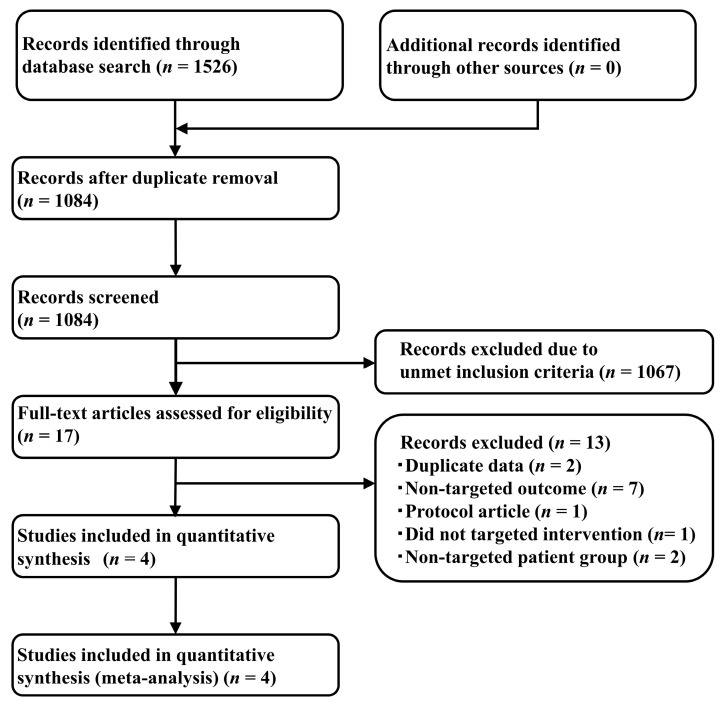
Study selection process.

**Figure 2 cancers-13-00052-f002:**
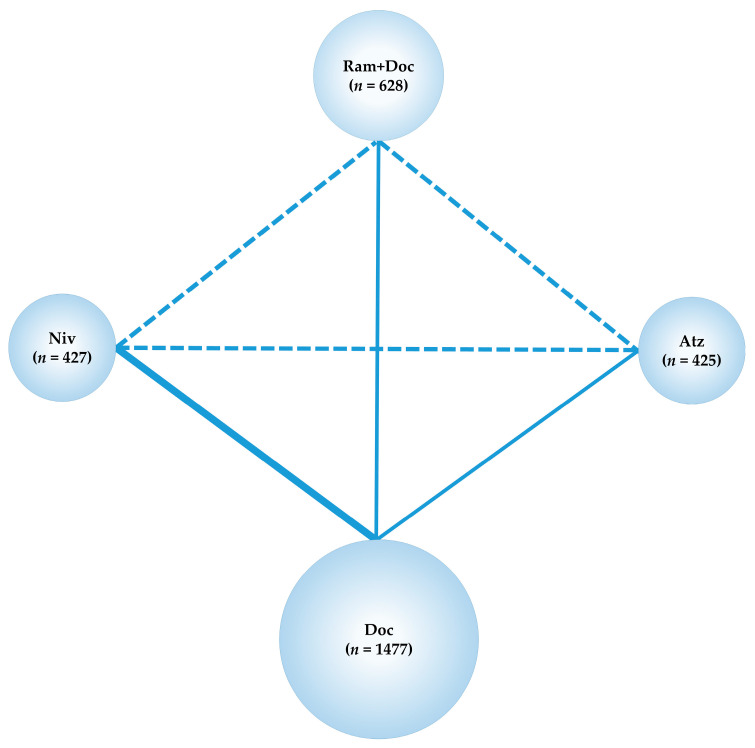
Network map. Solid lines represent randomized controlled trials (RCTs) while relative thickness represents the number of included studies. The dashed line reveals the absence of RCTs, suggesting that an indirect treatment comparison could be attempted. Circle size reflects the proportion of patients included in each treatment group. Ram+Doc (ramucirumab plus docetaxel); Niv (nivolumab); Atz (atezolizumab); *n*, number of patients.

**Figure 3 cancers-13-00052-f003:**
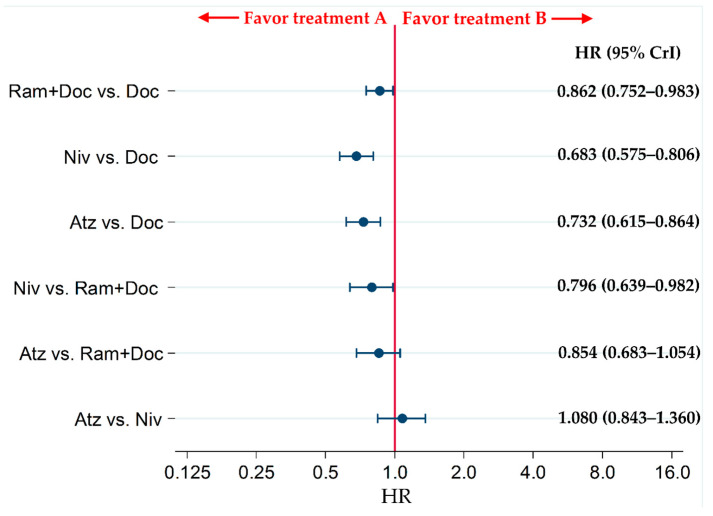
Comparative overall survival (OS) of patients with refractory or relapsed advanced non-small-cell lung cancer in the Ram+Doc, Doc, Niv and Atz groups. Patient inclusion criteria for all RCTs included in this analysis did not place any limitations on PD-L1 status. Comparisons are represented as treatment A versus treatment B. Data are presented as hazard ratios (HR) with 95% credible intervals (CrI). Ram+Doc (ramucirumab plus docetaxel); Doc (docetaxel); Niv (nivolumab); Atz (atezolizumab).

**Figure 4 cancers-13-00052-f004:**
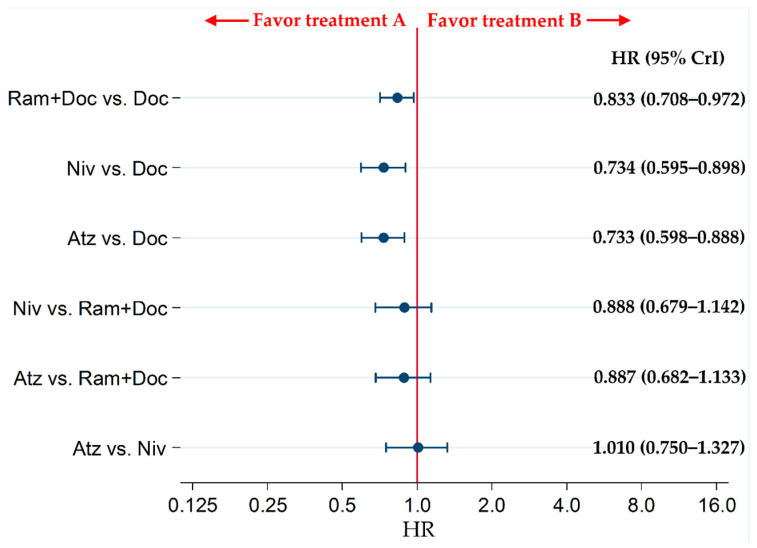
Comparing the OS of patients with refractory or relapsed advanced non-squamous non-small-cell lung cancer in the Ram+Doc, Doc, Niv and Atz groups. The patient inclusion criteria for all RCTs included in this analysis did not place any limitations on PD-L1 status. Comparisons are represented as treatment A versus treatment B. Data are presented as HRs (95% CrIs). Ram+Doc, (ramucirumab plus docetaxel); Doc (docetaxel); Niv (nivolumab); Atz (atezolizumab); HR (hazard ratio); CrI (credible interval).

**Figure 5 cancers-13-00052-f005:**
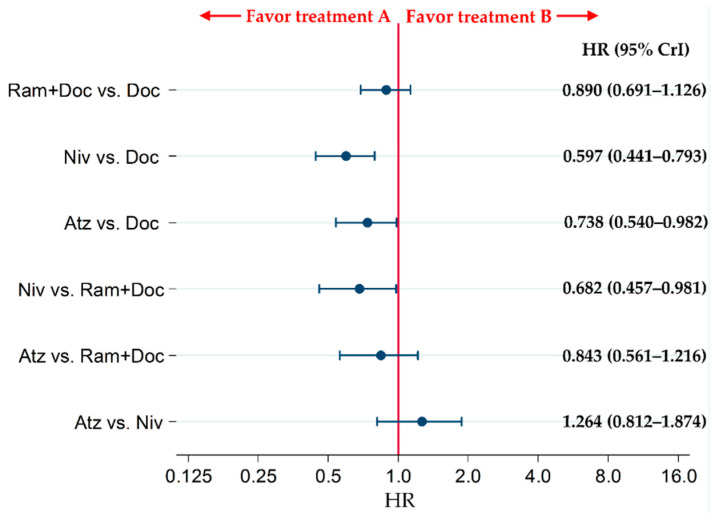
Comparative efficacy of the four therapeutic regimens, Ram+Doc, Doc, Niv and Atz, pertaining to the OS of patients with refractory or relapsed advanced squamous non-small-cell lung cancer. The patient inclusion criteria for all RCTs in this analysis did not place any limitations on PD-L1 status. Comparisons are represented as treatment A versus treatment B. Data are presented as HRs and 95% CrIs. Ram+Doc (ramucirumab plus docetaxel); Doc (docetaxel); Niv (nivolumab); Atz (atezolizumab); HR (hazard ratio); CrI (credible interval).

**Figure 6 cancers-13-00052-f006:**
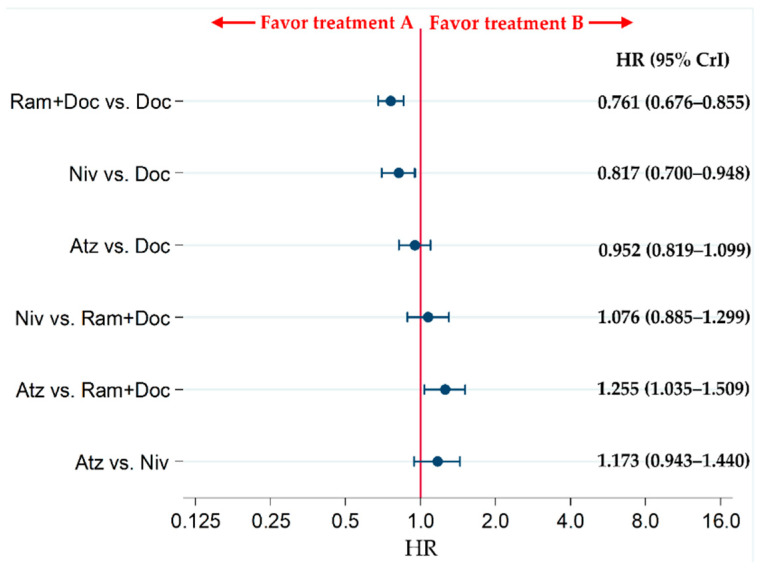
Comparing the progression-free survival (PFS) of patients with refractory or relapsed advanced non-small-cell lung cancer in the Ram+Doc, Doc, Niv, and Atz groups. The patient inclusion criteria for all RCTs in this analysis did not place any limitations on PD-L1 status. Comparisons are represented as treatment A versus treatment B. Data are presented as HRs (95% CrIs). Ram+Doc (ramucirumab plus docetaxel); Doc (docetaxel); Niv (nivolumab); Atz (atezolizumab); HR (hazard ratio); CrI (credible interval).

**Figure 7 cancers-13-00052-f007:**
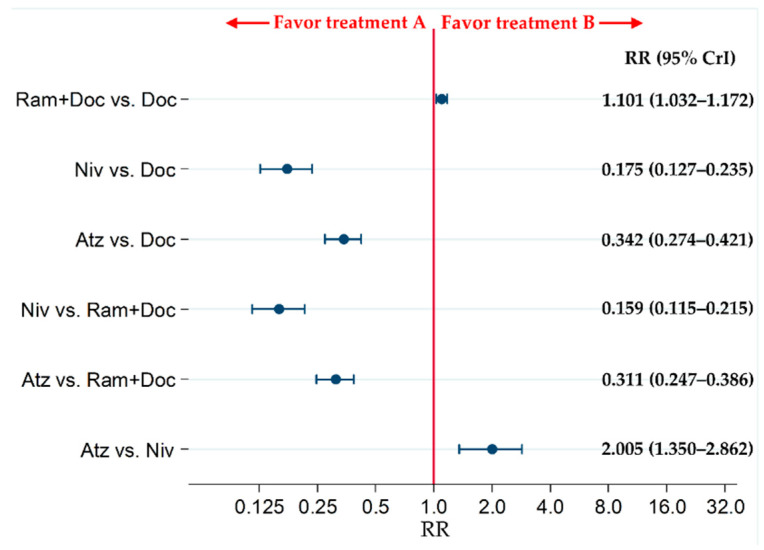
Comparative safety of the four therapeutic regimens, Ram+Doc, Doc, Niv, and Atz, pertaining to grade 3–5 treatment-related adverse events in patients with refractory or relapsed advanced non-small-cell lung cancer. The patient inclusion criteria for all RCTs in this analysis did not place any limitations on PD-L1 status. Comparisons are expressed as treatment A versus treatment B. Data are presented as RRs and 95% CrIs. Ram+Doc (ramucirumab plus docetaxel); Niv (nivolumab); Atz (atezolizumab); Doc (docetaxel); RR (risk ratio); CrI (credible interval).

**Figure 8 cancers-13-00052-f008:**
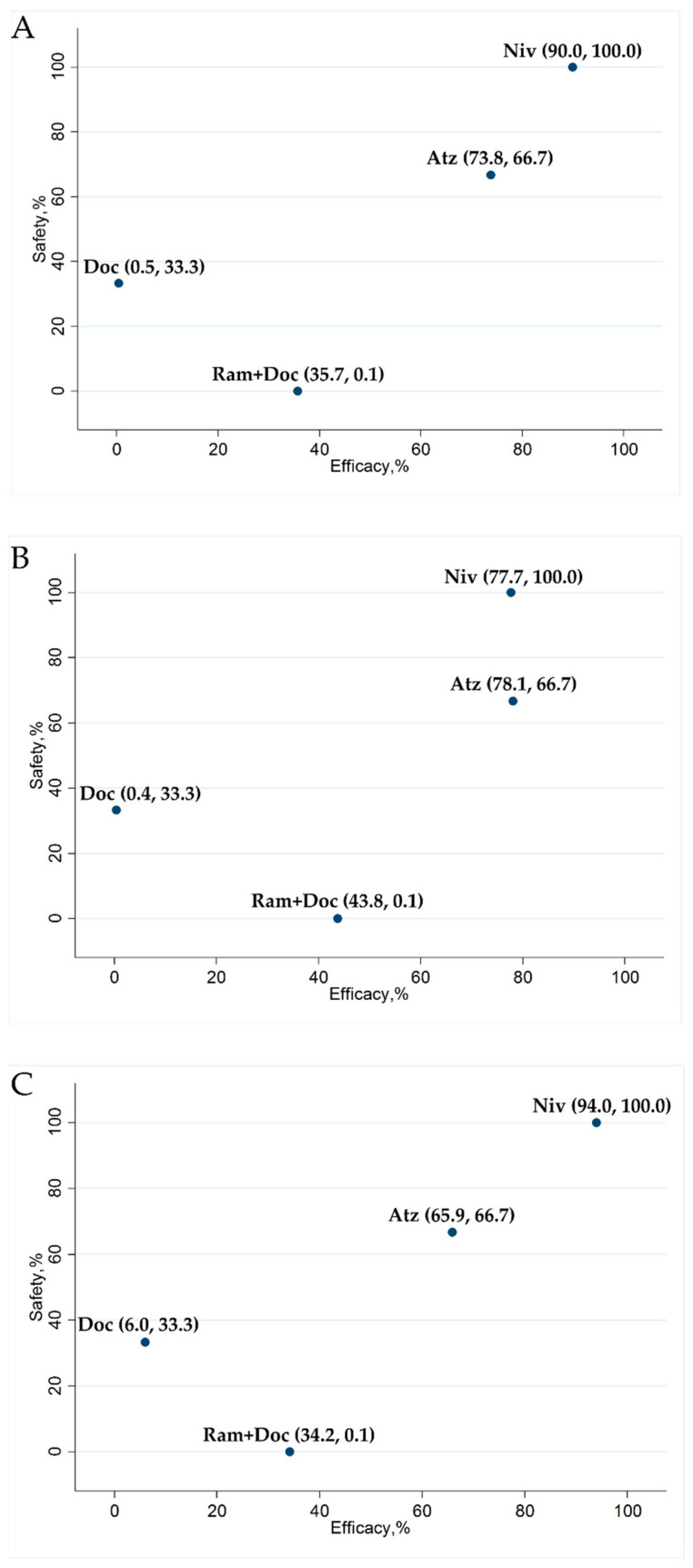
Scatter diagrams of SUCRAs for OS and safety in terms of G3–5AEs among Ram+Doc, Niv, Atz and Doc treatments in refractory or relapsed advanced NSCLC patients (**A**) OS in the overall population was highest following Niv treatment, followed by Atz, Ram+Doc and Doc; (**B**) In a subgroup analysis of OS in patients with non-squamous NSCLC, Atz treatment was associated with the highest OS, followed by Niv, Ram+Doc, and Doc; (**C**) In a subgroup analysis of OS in squamous NSCLC patients, Niv treatment was associated with the highest OS, followed by Atz, Ram+Doc, and Doc. Meanwhile, Niv treatment ranked the highest in terms of safety, followed by Atz, Doc, and Ram+Doc (**A**–**C**). SUCRA (surface under the cumulative ranking area curve); G3–5AEs (grade 3–5 treatment-related adverse events); Niv (nivolumab); Atz (atezolizumab); Ram+Doc (ramucirumab plus docetaxel); Doc (docetaxel).

**Figure 9 cancers-13-00052-f009:**
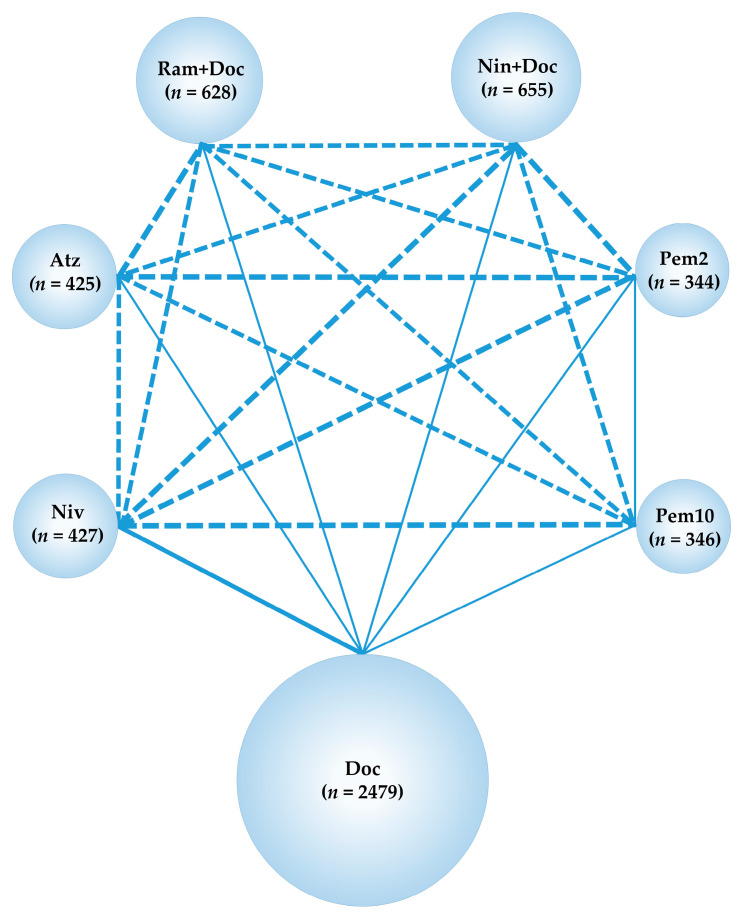
Network map of the analysis comparing the efficacy of the seven treatments, Doc, Ram+Doc, Niv, Atz, Pem2, Pem10, and Nin+Doc on the overall survival of participants of six RCTs, REVEL, CheckMate057, CheckMate017, OAK, KEYNOTE-010, and LUME-Lung 1. Solid lines represent randomized controlled trials (RCTs), and relative thickness represents the number of included studies. The dashed line reveals the absence of RCTs, suggesting that an indirect treatment comparison could be attempted. Circle size reflects the proportion of patients included in each treatment group. Ram+Doc, ramucirumab plus docetaxel; Niv, nivolumab; Atz, atezolizumab; Pem2, pembrolizumab 2 mg/kg; Pem10, pembrolizumab 10 mg/kg; *n*, number of patients.

**Figure 10 cancers-13-00052-f010:**
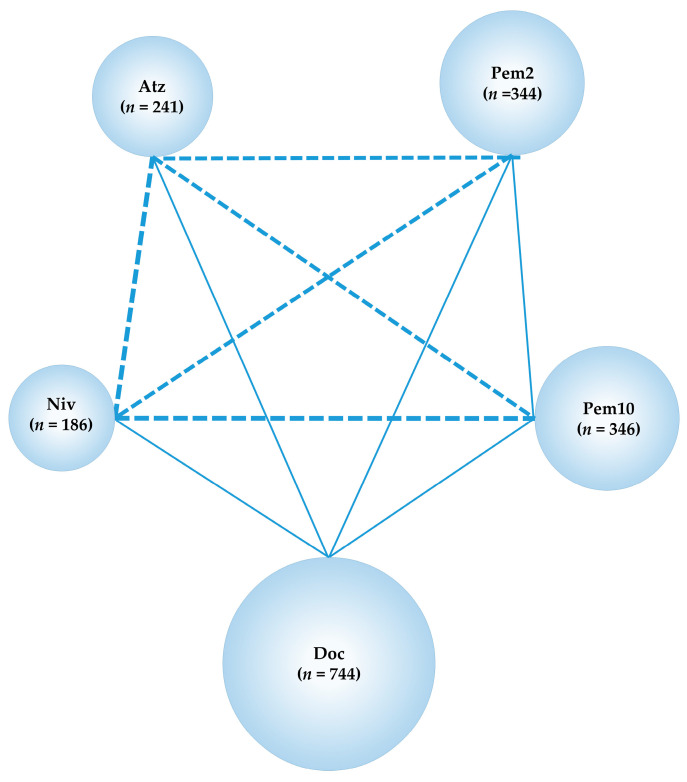
Network map of subgroup analysis for refractory or relapse PD-L1-positive (≥1%) advanced NSCLC. Solid lines represent randomized controlled trials (RCTs), and relative thickness represents the number of included studies. The dashed line reveals the absence of RCTs, suggesting that an indirect treatment comparison could be attempted. Circle size reflects the proportion of patients included in each treatment group. Ram+Doc, ramucirumab plus docetaxel; Niv, nivolumab; Atz, atezolizumab; Pem2, pembrolizumab 2 mg/kg; Pem10, pembrolizumab 10 mg/kg; n, number of patients.

**Figure 11 cancers-13-00052-f011:**
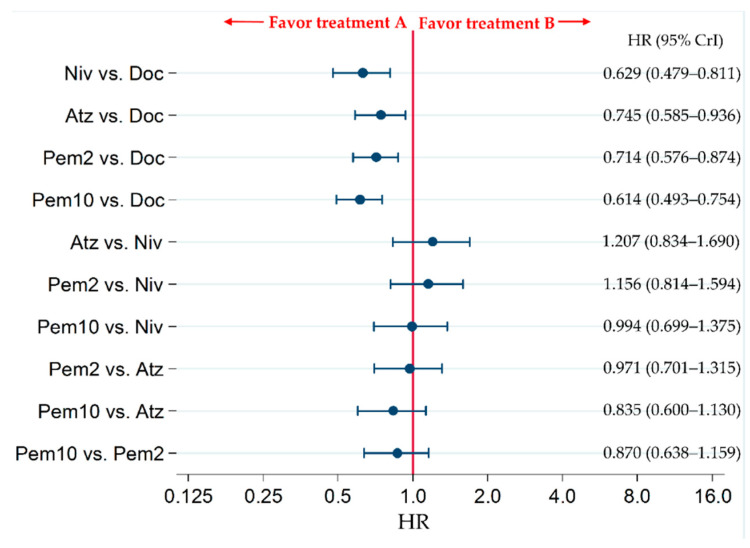
Comparative overall survival (OS) of patients with programmed cell death legend 1 (PD-L1) positive (≥1%) refractory or relapsed advanced non-small-cell lung cancer in the Doc, Niv, Atz, Pem2, and Pem10 groups. Comparisons are represented as treatment A versus treatment B. Data are presented as hazard ratios (HR) with 95% credible intervals (CrI); Doc (docetaxel); Niv (nivolumab); Atz (atezolizumab); Pem2 (pembrolizumab 2 mg/kg); Pem10 (pembrolizumab 10 mg/kg).

**Figure 12 cancers-13-00052-f012:**
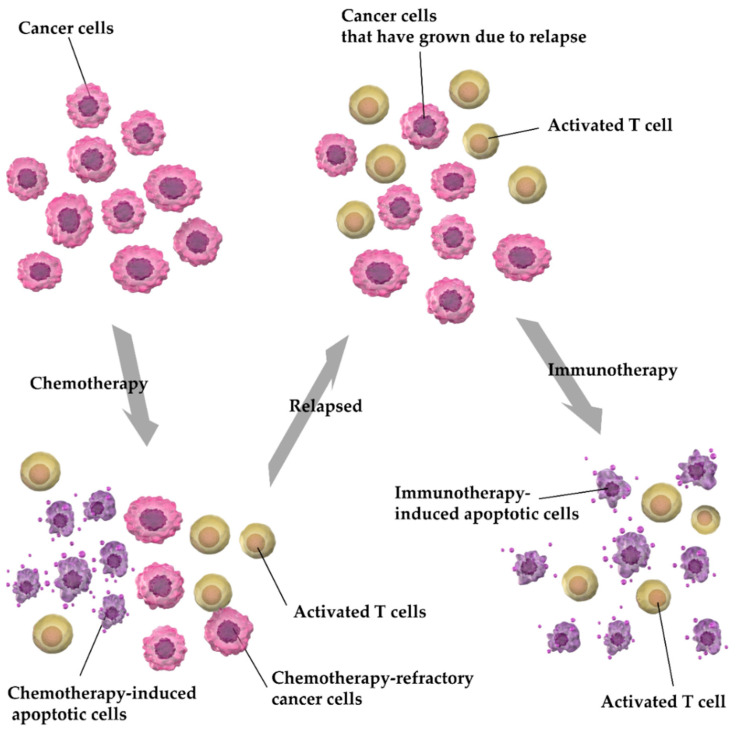
Mechanisms contributing to the efficacy of immunotherapy in refractory and relapsed cases. Initial treatments, such as chemotherapy, eliminate cancer cells but treatment-resistant cancer cells may survive. Treatment induces inflammation in major vicinities, leading to the formation of active T cells. The antitumor effects of these activated T cells are suppressed by cancer cells. However, immunotherapy reactivates T cells, thereby causing an anticancer effect in tissues with recurrent cancer.

**Table 1 cancers-13-00052-t001:** Key criteria for study inclusion.

Study Names	Key Inclusion Criteria
REVEL [9]	Age ≥ 18 years
	Stage IV squamous or non-squamous NSCLC
	Progressed during or following a single platinum-based chemotherapy regimen
	Performance status score of 0 or 1
CheckMate057 [28]	Age ≥18 years
	Stage IIIB or IV or recurrent non-squamous NSCLC
	Recurrence or progression during or following one prior platinum-based doublet chemotherapy
	Performance status score of 0 or 1
CheckMate017 [27]	Age ≥ 18 years
	Stage IIIB or IV squamous-cell NSCLC
	Disease recurrence following previous treatment with a platinum-containing regimen
	Performance status score of 0 or 1
OAK [29]	Age ≥ 18 years
	Stage IIIB or IV squamous or non-squamous NSCLC
	Previously received 1–2 cytotoxic chemotherapy regimens (≥1 platinum-based combination therapy)
	Performance status score of 0 or 1

NSCLC, non-small-cell lung cancer.

**Table 2 cancers-13-00052-t002:** Characteristics of the included studies.

Study Names	Treatment Arms	*N*	Age, Year Median (Range)	Females No. (%)	ECOG PS No. (%)	Histological Type No. (%)
REVEL [9]	Ramucirumab (10 mg/kg) plus docetaxel (75 mg/m^2^) on day 1 of 21-day cycle	628	62 (21–85)	209 (33)	PS0: 207 (33)	Non-squamous: 465 (74)
					PS1: 420 (67)	Squamous: 157 (25)
						Unknown: 6 (1)
	Placebo plus docetaxel (75 mg/m^2^)	625	61 (25–86)	210 (34)	PS0: 199 (32)	Non-squamous: 447 (72)
	on day 1 of 21-day cycle				PS1: 425 (68)	Squamous: 171 (27)
						Unknown: 7 (1)
		Total: 1253				
CheckMate057 [28]	Nivolumab (3 mg/kg e2w)	292	61 (37–84)	141 (48)	PS0: 84 (29)	Non-squamous: 292 (100)
					PS1: 208 (71)	Squamous: 0 (0)
					NR: 0	
	Docetaxel (75 mg/m^2^ e3w)	290	64 (21–85)	122 (42)	PS0: 95 (33)	Non-squamous: 290 (100)
					PS1: 194 (67)	Squamous: 0 (0)
					NR: 1 (<1)	
		Total: 582				
CheckMate017 [27]	Nivolumab (3 mg/kg e2w)	135	62 (39–85)	24 (18)	PS0: 27 (20)	Non-squamous: 0 (0)
					PS1: 106 (79)	Squamous: 135 (100)
					NR: 2 (1)	
	Docetaxel (75 mg/m^2^ e3w)	137	64 (42–84)	40 (29)	PS0: 37 (27)	Non-squamous: 0 (0)
					PS1: 100 (73)	Squamous: 137 (100)
					NR: 0 (0)	
		Total: 272				
OAK [29]	Atezolizumab (1200 mg e3w)	425	63.0 (33.0–82.0)	164 (39)	PS0: 155 (36)	Non-squamous: 313 (74)
					PS1: 270 (64)	Squamous: 112 (26)
	Docetaxel (75 mg/m^2^ e3w)	425	64.0 (34.0–85.0)	166 (39)	PS0: 160 (38)	Non-squamous: 315 (74)
					PS1: 265 (62)	Squamous: 110 (26)
		Total: 850				

N, sample size; ECOG, Eastern Cooperative Oncology Group; PS, performance status; e3w, every 3 weeks; e2w, every 2 weeks; NR, not reported.

## Data Availability

The authors confirm that the datasets analyzed during the current study are available from the corresponding author upon reasonable request.

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
