# Peer review of "Comparative Efficacy and Safety of Anti-PD-1/PD-L1 Immune Checkpoint Inhibitors for Refractory or Relapsed Advanced Non-Small-Cell Lung Cancer—A Systematic Review and Network Meta-Analysis"

_cancers, 2020, doi:10.3390/cancers13010052_

Round 1
Reviewer 1 Report
The Metaanalysis addresses the relevant comparison of 2L IO vs. Doce/Ramucirumab. This is clinically relevant, since generally Doce mono without Ramu was used as a comparator in the 2L IO-Phase III studies.
A drawback is the fact that 2L pembrolizumab could not be analyzed due to the restriction on PD-L1 pos. patients. However, this cannot be changed for statistical reasons.
The results drawn from the metaanalysis both with respect to efficacy and safety seem well founded and are in line with clinical experience.
Therefore, this metaanalysis should be accepted for publication in Cancers.
Reviewer 2 Report
The authors have adequately addressed all reviewers comments. I would like to recommend to accept this paper in the current form.This manuscript is a resubmission of an earlier submission. The following is a list of the peer review reports and author responses from that submission.
Round 1
Reviewer 1 Report
The authors provide a well written comparison of 2L ICIs or ramucirumab + doce vs. doce. This is an interesting comparison of clinical relevance. Of particular interest is the indirect comparison of two ICIs which should be supplemented by pembrolizumab as the third major ICI approved for 2L use in NSCLC. Furthermore, the comparison of RAM and nintedanib as antiangiogenic agents should be added for patients with adenocarcinoma.
The relevance is limited by the fact that ICIs are nowadays generally used in 1L. Therefore the analysis applies only to a minority of NSCLC patients treated according to current guidelines.
Major:
Pembrolizumab is approved for 2nd line treatment of PD-L1+ NSCLC and needs to be included in the analysis of PD-L1+ patients (for comparison with ATZ and NIVO as subgroup analysis of PD-L1+ patients, for comparison with doce/Ram/doce without PD-L1 constraint). If separate PD-L1+ data are not available for ATZ/NIVO, then the Pembro data (KN 010) need to be discussed (comparison of efficacy (HR) and safety with those for ATZ/NIVO).
Since subgroup analysis for non-squamous NSCLC are available for all 4 studies in this analysis, the LUME-lung 1 data should be included in the subgroup analysis.
In the discussion, the authors need to emphasize that their analysis is only relevant for patients not treated with IO-chemo combinations which represent the current standard of care.
Minor:
L54: although recent publications are quoted, it should be stated that these OS-data are based mainly on patients not treated with IO.
L62: Nintedanib + doce needs to be mentioned as another the option for adenocarcinoma (Reck, LUME-Lung 1, Lancet Oncol. 2014).
L63: the term “targeted” therapies should be reserved for small molecule targeted therapies such as EGFR-TKIs to distinguish these therapies from ICIs.
L74: ref. 13-15 refer to papers describing statistical methods published before use of ICIs. The text, however, suggests that these papers describe comparisons of ICIs. This needs clarification.
L102: ref. of the Pembrol. 2nd line study is needed (KEYNOTE 010, Herbst, Lancet 2016).
Reviewer 2 Report
Your article doesn't reflect the current clinical standard in first and second line setting in NSCLC. The first line situation has been changed to chemotherapy PLUS immunotherapy. You have mentioned it neither in your introduction nor in your discussion. So far, the clinical relevance of your meta-analysis is not really interesting for the majority of the clinicians. Your analysis included no data on second line immunotherapy versus docetaxel or docetaxel+ramucirumab after first line treatment with chemotherapy PLUS immunotherapy.
Your article is lacking data on PD-L1 expressions. As you surely know, PD-L1 status has an impact on OS. You even did not discuss this issue.
Furthermore, data on pembrolizumab are lacking. Pembrolizumab is a very good established IO drug using by higher number of oncologists.